# Hopping Conductivity and Dielectric Relaxations in Ag/PAN Nanocomposites

**DOI:** 10.3390/polym13193251

**Published:** 2021-09-24

**Authors:** M.A. Kudryashov, A.A. Logunov, L.A. Mochalov, Yu.P. Kudryashova, M.M. Trubyanov, A.V. Barykin, I.V. Vorotyntsev

**Affiliations:** 1Department of Chemical Technology, Nizhny Novgorod State Technical University, n.a. R.E. Alekseev, 603950 Nizhny Novgorod, Russia; kudryashov@phys.unn.ru (M.A.K.); alchemlog@gmail.com (A.A.L.); kudryashova@phys.unn.ru (Y.P.K.); irflex@yandex.ru (M.M.T.); barykin@gmail.com (A.V.B.); ilyavorotyntsev@gmail.com (I.V.V.); 2University of North Carolina at Charlotte, Charlotte, NC 28223, USA; 3Sirius University of Science and Technology, 354340 Sochi, Russia; 4Department of Membrane Technology, Mendeleev University of Chemical Technology of Russia, Miusskaya Sq. 9, 125047 Moscow, Russia

**Keywords:** polymer nanocomposites, silver nanoparticles, electrical modulus, hopping conductivity

## Abstract

The dependence of the conductivity and electric modulus of silver/polyacrylonitrile nanocomposites on the frequency of an alternating electric field has been studied at different temperatures and starting mixture AgNO_3_ contents. The frequency dependences on the conductivity of the nanocomposites in the range of 10^3^–10^6^ Hz are in good agreement with the power law *f*^0.8^. The observed relaxation maxima in the relation of the imaginary part of the electric modulus on the frequency can be explained by interfacial polarization. It was shown that the frequency dispersions of conductivity and electric modulus were well described by the Dyre and Cole-Davidson models, respectively. Using these models, we have estimated the relaxation times and the activation energies of these structures. A mechanism of charge transport responsible for the conductivity of nanocomposites is proposed. An assumption is made regarding the presence of Ag_4_^2+^ and Ag_8_^2+^ silver clusters in the polymer.

## 1. Introduction

Nanocomposites including dispersed conductive nanoparticles in a dielectric matrix have attracted considerable interest from researchers due to their possible electrical and electromagnetic applications [1]. Such uses include, for example, the screening of electromagnetic or radio interference [2] and electrostatic charge dissipation [3]. Metal–polymer nanocomposites can be used as electrically conductive adhesives and circuit elements in microelectronics [4], strain and pressure sensors [5], flexible tactile devices [6], and gas analyzers [7]. They also possess anti-corrosion properties and may be used as a coating for metal contacts [8]. The manufacture of microelectronic components involves exposure at high temperatures, which can lead to cracking, chipping, or diffusion of even very small amounts of metal through the polymer into the semiconductor. Therefore, a great deal of effort has been undertaken to improve control of the microstructure and thermal stability of metal–polymer interfaces in order to prevent their degradation and improve adhesion [9,10,11]. In particular, it was found that in polymer composites containing metal nanoparticles, the permittivity is sufficiently high to enable the use of such materials in electronics and microwave technology. For all of the applications mentioned above, as a rule, solid continuous metal–polymer nanocomposites are required. To reduce their cost, the preparation conditions must include low temperatures, sufficiently high synthesis rates, and the technology must be scalable.

In order to study the electrical properties of such systems, they are considered to be heterogeneous, and various relationships are used within the framework of the theories of effective media, describing their behavior based on the dielectric constants and specific electrical conductivities of the constituent parts [12]. Electrical characteristics of metal–polymer nanocomposites are related to the volume fraction, size, and shape of the metal particles. For example, the use of pointed metal particles in composites significantly improves electrical conductivity, since the charges accumulated at the tips of these particles generate a very large local electric field [13], which increases the probability of tunneling through the insulating barrier. In this case, electrons tunnel between sharp tips at greater distances than in the case of spherical particles, which leads to a more significant decrease in electrical resistance. In [5,14], the influence of the morphological parameters (radius of curvature, tip height, etc.) of such pointed particles on the resistance of piezoresistive composites was demonstrated. It was found that needle-like particles with a small core size and a small radius of curvature of the tips provide a strong enhancement of tunneling conductivity, and the synthesized gold nanostars show very good characteristics in terms of tunneling conductivity at a low concentration in composites. Dielectric relaxations in metal–polymer composites can be studied using dielectric spectroscopy. However, in the study of such materials, where the dielectric constant at low frequencies of the electric field can reach values of more than 1000, there is a problem of detecting and identifying relaxations. As a matter of fact, in this case, the relaxations were hidden due to the presence of electrically conductive inclusions in the dielectric matrix. Therefore, to identify them, the inverse value of the complex dielectric constant was used—the electrical modulus [15,16].

In this study, the frequency dependences of conductivity and the electric modulus of silver/polyacrylonitrile (Ag/PAN) nanocomposites produced using simultaneous acrylonitrile polymerization and silver ion reduction by UV-radiation were examined. This method makes it possible to obtain nanocomposites at low temperatures, is characterized by high rates of polymerization, and results in the formation of uniformly dispersed metal nanoparticles with a small spread in size during the growth of the polymer net. In turn, this makes the production of films easier, and the particle size is well controlled within the required mode. UV polymerization is widespread in industry and is most suitable in terms of the parameters of obtaining cheap nanocomposites of large areas and in huge quantities. PAN is an important polymer used primarily in the manufacture of artificial fibers, as well as for the production of ultrafiltration membranes. Silver, as a well-known metal, is characterized by high electrical conductivity, amazing optical properties, and interesting oxidative properties in catalysis. It is silver nanoparticles, in contrast to copper and gold, that make it possible to regulate the plasmon resonance band in the entire visible range and in the near-IR region. In addition, silver nanoparticles scatter light and enhance the near fields by an order of magnitude more than in the case of gold [17]. On the basis of silver/polymer nanocomposites, effective antibacterial materials and devices can be created [18,19].

## 2. Experimental Section

We studied the UV-polymerization of silver nitrate (Sigma-Aldrich, Saint Louis, MO, USA) in acrylonitrile (Sigma-Aldrich, Saint Louis, MO, USA) in the presence of 15 wt.% 2,2-dimethoxy-2-phenylacetophenone (Sigma-Aldrich, Saint Louis, MO, USA) as a photoinitiator, producing Ag/PAN nanocomposite films (PI). After mixing the monomer, PI, and AgNO_3_, the mixture was placed between two glass plates with a conductive ITO layer with a size of 3 × 1.5 cm^2^ and a 20 μm gap between them, due to capillary effect, and polymerized for 90 min with collimated UV light (*λ* = 365 nm and 155 μW/cm^2^). Thus, the sample presented itself as a flat capacitor with a nanocomposite film between the plates. In separate experiments, AgNO_3_ concentrations were varied from 0 to 30 wt.%. A more detailed description of the synthesis of nanocomposites is provided in [20,21].

To study their morphology, the obtained nanocomposite films were separated from the glass and placed in silicone cells with epoxy resin in the shape of a rectangular parallelepiped, which were then heated in an oven at 60 °C for 24 h for curing. From the finished solid parallelepipeds on a Leica UC7 (ultramicrotome, Leica Microsystems, Wetzlar, Germany), transverse sections with a thickness of about 100 nm were cut, which were placed on copper grids for electron microscopy. The morphology of ultrathin cross-sections of the films was studied using a Morgagni 268D (transmission electron microscope, FEI, Hillsboro, Oregon) with a magnification of ×2.8×10^5^. The geometric characteristics of silver nanoparticles were determined using the SPMLab ™ v5 and Gwyddion v2.58 software.

The HP 4284A (LCR-meter, Hewlett-Packard, Palo Alto, CA, USA) was used to measure conductivity and capacitance versus the frequency of the electric field in the equivalent circuit with the resistor and capacitor connected in parallel in the frequency range of 20–10^6^ Hz. Using the LOIP LT-100 (circulation thermostat, LOIP, St. Petersburg, Russia) with external cooling, the sample was maintained at a temperature in range of 10–60 °C. An E6-13A (teraohmmeter, Punane-Rat, Estonia, Tallinn) was used to test the DC resistance. Based on the capacitance and conductance measurements of these structures, the real and imaginary components of the dielectric constant were calculated.

## 3. Results and Discussion

The TEM images presented in Figure 1 show the dependence of the size and density of the formed silver nanoparticles on the AgNO_3_ content in the initial reaction mixture. Nanoparticles are quite uniformly distributed in the volume of the polymer matrix, and their shape is close to spherical. The particle size distribution (Figure 1d–f) is described by a Gaussian with a narrow symmetric maximum, which indicates a small spread of silver nanoinclusions in size for such systems. PAN acts as a stabilizer for particles and prevents their agglomeration. The size of Ag nanoparticles increases as the amount of silver nitrate in the initial mixture increases. In the case of 5 wt.% AgNO_3_ (not shown here), the average nanoparticle diameter is about 3.5 nm and increases to 13 nm at 30 wt.%. Note that a growth in the amount of metal salt in the starting mixture leads to an increase in the volume fraction of silver in PAN due to the formation of a larger number of nuclei.

### 3.1. Dielectric Dispersion

It is well known and has been validated in [22] that there are dielectric relaxations in polymer matrix-based composites. Interfacial polarization (the Maxwell–Wagner effect) occurs in metal–polymer composites due to the presence of free charge carriers in the system, which migrate under the influence of an applied field and accumulate at the interface between media with significantly different permittivity and conductivity. Large dipoles arise on the surface of metal particles as a result of this. The conductivity and permittivity of the composite material components determine this type of relaxation. Because of the dipoles’ inertia, such relaxation occurs at low frequencies and is the slowest of all dielectric processes.

When the electric modulus is utilized, the changes in the broad range of permittivity and conductivity at low frequencies are minimized, providing an advantage in interfacial polarization interpretation. The normal challenges associated with the influence of the electrode nature, contact ohmicity, and the effects of space charge injection on the frequency curves of the real and imaginary components of the dielectric constant that “mask” the relaxation can be resolved or even ignored in this regard [23]. The electric modulus *M** is defined by the following equation:(1)$$M^{*}=\frac{1}{\varepsilon^{*}}=\frac{1}{\varepsilon^{\prime }-j\varepsilon^{\prime \prime }}=\frac{\varepsilon^{\prime }}{\varepsilon^{{}^{\prime }2}+\varepsilon^{{}^{\prime \prime }2}}+j\frac{\varepsilon^{\prime \prime }}{\varepsilon^{{}^{\prime }2}+\varepsilon^{{}^{\prime \prime }2}}=M^{\prime }+jM^{\prime \prime }$$
where *M*′ and *M*″ are the real and the imaginary parts of the electric modulus, and *ε*′ and *ε*″ are the real and the imaginary parts of permittivity, respectively.

The frequency dependences of the electric modulus of our composites (obtained using Equation (1)) are shown in Figure 2 and Figure 3. The real part of the electric modulus (*M*′) decreases as the amount of silver nitrate in the initial mixture increases, and the temperature increases as the real element of permittivity increases. Other composite materials with conductive inclusions have shown similar behavior [24,25,26]. The relaxation process is assumed by an abrupt shift from low to high values, which must be followed by maximum losses in the dependences of the imaginary component of the electric modulus (*M*″) on frequency. This may be shown in Figure 2 and Figure 3.

As the temperature increases, the relaxation peaks shift to higher frequencies. At the same time as the silver content in the polymer increases, the maximum tends to decrease at the constant measurement temperature. This effect is based on interfacial polarization, which is supported by the theory published in [27,28] and prior experimental results in similar systems [25,29].

The shift in the maxima to higher frequencies when the volume fraction of silver in polyacrylonitrile (AgNO_3_ concentration in the initial mixture) increases could be attributed to an increase in the conductivity of individual metal nanoparticles [27,28]. As previously demonstrated, increasing the concentration of AgNO_3_ causes the size of the produced silver nanoparticles to rise. This in turn may result in the increase of electrical conductivity in these nanoparticles, which (up to a certain size of nanoinclusion) differs from the conductivity of the bulk material.

The electrical modulus frequency dependences are inconsistent with the fundamental Debye model [30]. In comparison to the solely Debye relaxation process, the frequency dependence maxima of *M*″ is wider and lower. The Maxwell–Wagner–Sillars equations [31] based on a simple Debye relaxation also result in more narrow and abrupt peaks, overestimating the electric modulus (*M*″) loss coefficient [32]. The Debye model and the Maxwell–Wagner–Sillars equations correspond to a single relaxation time process, which does not appear to be the case with the nanocomposites studied in this article. To describe the dielectric properties of polymer systems, the models of Cole–Cole [33], Cole–Davidson [34], Havriliak–Negami [35], and Kohlrausch–Williams–Watts [36] are often used. All of these approaches involve some distribution of relaxation times. By analogy with [24,25], in order to explain the dependence of the electrical modulus on the frequency we used the Cole–Davidson model, according to which *M′* and *M*″ are as follows:(2)$$M^{\prime }=\frac{M_{\infty }M_{s}\left[M_{s}+\left(M_{\infty }-M_{s}\right){\left(\cos\varphi \right)}^{\gamma }\cos\gamma \varphi \right]}{M_{s}^{2}+\left(M_{\infty }-M_{s}\right){\left(\cos\varphi \right)}^{\gamma }\left[2M_{s}\cos\gamma \varphi +\left(M_{\infty }-M_{s}\right){\left(\cos\varphi \right)}^{\gamma }\right]}$$
(3)$$M^{\prime \prime }=\frac{M_{\infty }M_{s}\left(M_{\infty }-M_{s}\right){\left(\cos\varphi \right)}^{\gamma }\sin\gamma \varphi }{M_{s}^{2}+\left(M_{\infty }-M_{s}\right){\left(\cos\varphi \right)}^{\gamma }\left[2M_{s}\cos\gamma \varphi +\left(M_{\infty }-M_{s}\right){\left(\cos\varphi \right)}^{\gamma }\right]}$$
where *M_s_* and *M*_∞_ are the values of *M*′ when *ω* → 0 and ω → ∞, respectively,
(4)$$0<\gamma \leq 1,\;\mathrm{tg}\varphi =\omega \tau ,\;\omega_{max}\tau =\mathrm{tg}\left(\frac{1}{\gamma +1}\frac{\pi }{2}\right),$$
where *ω_max_* is the frequency of maximum loss on the curve *ε*″(*f*) (*ω_max_* = 2 *π f_ε_*_,*max*_), and *τ* is the relaxation time associated with the static electric field (often referred to as *τ_ε_*). The relaxation time associated with the constant displacement vector is defined as *τ**_M_* = (*M_s_*/*M**_∞_*)*τ_ε_*, and the dislocation of the maximum on the *M″* curve as *f_M, max_* = (*M**_∞_*/*M_s_*) *f**_ε,max_* [32]. The value of *γ* = 1, as a measure of the relaxation time width, corresponds to one moment of relaxation or Debye relaxation.

Figure 4 shows the dependences of the imaginary part of the electric modulus on the real part (the Cole–Cole equation). Condensed half-circles correspond to relaxation processes occurring in each of the samples.

Almost all of the Cole–Davidson model’s experimental data and approximation curves pass through the origin. This shows that in the systems under investigation, there is no additional relaxing process at lower frequencies. The impact of the silver content is reflected in the changes in the radius of half circles.

The experimental data corresponds well with the Cole–Davidson model (Figure 2, Figure 3 and Figure 4). Parameters for fitting were *γ* and *τ_M_*. The γ values were all greater than 0.59, indicating a relatively narrow relaxation time distribution. As the silver content of the composites increased, *γ* tended to climb, implying that relaxation was approaching a purely Debye process. Because the wasted thermal energy helps the movement of the dipoles created in the alternating electric field, the relaxation times for all systems decreased as the temperature rose. Furthermore, as the volume fraction of silver rises, the position of the falling maximum on the frequency plot of *M*″ changes to higher frequencies and the relaxation time decreases.

Figure 5 shows the dependences of the relaxation time on the inverse value of the temperature for the systems being studied. As can be seen, in Arrhenius coordinates, these dependencies are approximated well by straight lines, except for the nanocomposite prepared with 10 wt.% of AgNO_3_. In [25], the relaxation time is represented by the following expression:*τ* = *τ*_0_ exp (Δ*E*/*kT*),(5)
where Δ*E* is the activation energy of the relaxation process, *k* is the Boltzmann constant, and *T* is temperature. The values of Δ*E* obtained from the linear approximation and Equation (5), were 1.41 and 1.28 eV for the samples prepared with 20 and 30 wt.% of silver nitrate, respectively.

At high frequencies, experimental points deviated from the obtained curves, which is possibly due to the manifestation of different process of relaxation. This behavior was observed over the entire temperature range and for all silver/PAN nanocomposites. The nature of the dependence of the imaginary part of the electric modulus of nanocomposites in this frequency range is similar to the dependence of *M*″ for a pure polymer. The sharp increase in the imaginary part of the electrical modulus (Figure 1 and Figure 2) at high frequencies could have introduced the low-frequency edge of the loss maximum associated with the dipole polarization characteristic of polar polymers [27].

### 3.2. AC Conductivity

The filling factor of the metal and the size of the nanoinclusions have a big impact on the electrical characteristics of a polymer with distributed metallic nanoparticles. In metal–polymer nanocomposites, the following conduction processes are feasible in general:(1)ionic conductivity through the ions apportioned in the polymer matrix;(2)the polymer’s electronic conductivity;(3)electronic conductivity in the chain of contacting metal nanoparticles (metallic regime);(4)tunneling (hopping) electronic conductivity between isolated metal particles (dielectric regime).

In a dielectric matrix with large metal volume fractions, a sharp increase in conductivity occurs due to the formation of a three-dimensional conductive chain (percolation). It should be noted that a sharp increase in the conductivity of the metal–polymer composite is not always observed at metal volume fractions comparable to or higher than the expected percolation threshold. Such a case occurs when the polymer matrix tightly covers the surface of the metal particles; as a result, they do not make direct physical contact. As the content of the particles grows, this prevents particle chains from forming. As a result, even at filler quantities over the theoretical percolation threshold, the composite exhibits strong resistance. However, its conductivity increases sharply upon mechanical deformation, which makes it possible to use such the composite as a piezoresistive material [13,37,38,39]. In this case, the conductivity becomes very sensitive to the shape of the nanoparticles [14].

However, our structures have relatively low filling factors (up to 1%) and are never close to the percolation threshold [40,41,42]. Only at low frequencies does the ionic component contribute to the polymer’s conductivity. The conductivity of pure PAN and Ag/PAN nanocomposites is of the same order of magnitude at high frequencies (Figure 6a). Because the mobility of ions is substantially lower than that of electrons, this cannot be explained in terms of ionic conductivity. The presence of ions in our samples is only feasible due to the presence of impurities in the polymer or the dissociation of silver nitrate in the reaction mixture, because PAN is a covalent molecule.

The conductivity of pure PAN is proportional to *f* ^0.9^ in the frequency range below 2 × 10^4^ Hz, as seen in Figure 6a. According to [43], this type of frequency dependency of conductivity reveals a hopping mechanism of charge transport in the polymer. The conductivity of pure polymer is also regulated by the power law for frequencies over 2 × 10^4^ Hz, with the exponent p varying from 1.7 to 1.5 with increasing temperature. At frequencies over 10^6^ Hz, optical transitions [44] or absorption by a single-phonon acoustic mode enabled by disordering [45] should result in conductivity with *p* = 2. The conductivity of the nanocomposite film obtained at 2 wt.% AgNO_3_ was close to that of a pure polymer (not shown here).

The conductivity of Ag/PAN nanocomposites behaves similarly to that of pure polymers in the high-frequency range. A weak frequency dependency area is detected at low frequencies as the quantity of silver nitrate in the initial mixture increases, increasing the volume fraction of metal. This is also evident in Figure 6b, which depicts the curves without taking into account the conductivity of the polymer matrix. As a result, these curves only demonstrate conductivity in the presence of silver nanoparticles in the PAN. The results of the dependence are well represented by the power law *f*^0.8^, which is common for hopping conductivity [44,46] in the frequency range of 10^3^–10^6^ Hz. According to [25,47] the frequency conductivity of metal–polymer composites is represented by the formula:σ*_ac_* ≈ σ*_dc_* + *A ω^p^*,(6)
where *σ_dc_* is the DC conductivity, *ω* = 2*πf* is the angular frequency, and *A* and *p* are dependent on the temperature and the volume fraction of metal. Incidentally, *p* is constant and is approximately equal to 0.8. The field of low frequency dependence may be explained by the fact that in this field the DC conductivity is better than *A·ω*^0.8^ in this frequency range, and increases from 5.67 × 10^−7^ to 4.06 × 10^−4^ μSm∙cm^−1^ with the increase in the content of silver nitrate in the starting mixture from 2 to 30 wt.% of AgNO_3_.

Ag/PAN nanocomposite studies of frequency dependence of the conductivity at various temperatures (Figure 7) have shown that with the increase in sample temperature in the low and medium frequency regions, some “anomalous” dependence is observed in the curves, which cannot be described by Equation (6). This region is explained by the interfacial relaxation mentioned above. The AC conductivity temperature effect was more obvious in the low frequency zone, although the values of AC were relatively similar in the higher frequency field. The DC conductivity increased from 1.34 × 10^−5^ to 2.37 × 10^−3^ as the temperature rose from 285 to 333 K, respectively, due to which, as in the case of increased content of silver in the polymer, the portion of the curve where the law of *f*^0.8^ is fulfilled became smaller (Figure 6b).

In [25,47], the authors use the model of random potential barriers (also called a symmetric hopping model) proposed by Dyre [48] to define AC conductivity in metal composites. This model assumes non-interacting charge carriers, the hops of which are allowed only to the nearest “neighbor”. The jumping speed (the probability of jumping per unit of time) is considered symmetric, i.e., the same for jumping forward or backward between two localized states. It is assumed that the activation energy needed to overcome the barrier changes randomly. At low temperatures, the charge carrier is in the potential well most of the time. It may accidentally happen that it receives enough heat energy to jump into some nearby well, separated by a low or narrow potential barrier. In this case, the probability of the jump per unit of time is exp(*–*Δ*E/kT*), where *E_a_* is activation energy. According to this model, the complex value of AC conductivity can be expressed as:(7)$$\sigma_{ac}^{*}\left(\omega \right)=\sigma_{dc}\left[\frac{j\omega \tau }{\ln\left(1+j\omega \tau \right)}\right]$$
where *σ_dc_*, ω and *τ* are the DC conductivity, the angular frequency, and the relaxation time (mean jump time), respectively. Equation (7) ensures a fairly good definition of the frequency dependences of our nanocomposites’ conductivity (Figure 7). Relaxation time τ was used as a fitting parameter. The resulting relaxation times are consistent with the values of τM estimated from the Cole–Davidson model. They are well approximated by a straight line in the Arrhenius coordinates, except for the nanocomposite prepared with 10 wt.% of AgNO_3_. From the linear approximation and exponential Equation (5), we found activation energies of the relaxation process Δ*E*, which amounted to 1.03 and 0.96 eV for the samples made at 20 and 30 wt.% of silver nitrate, respectively. The observed correspondence confirms the applicability of the Dyre and Cole–Davidson models to describe the conductivity and electric modulus of Ag/PAN nanocomposites, respectively.

According to [49], the DC conductivity and the relaxation time we found can be related thus:*σ_dc_* = *p*Δεε_0_/*τ*,(8)
where Δ*ε* = *ε*(0) − *ε*(∞) and *p* is a temperature independent constant close to 1.

It follows from this expression that the relaxation time and DC conductivity have the same activation energy. The relations of the DC conductivity of silver/PAN nanocomposite films synthesized at 20 and 30 wt.% AgNO_3_ in the starting mixture on the measurement temperature in Arrhenius coordinates are shown in Figure 8. Obviously, the experimental points are also well approached by straight lines, as in the case of the relaxation time. The found activation energies for *σ_dc_* (1.28 and 1.15 eV for samples prepared at 20 and 30 wt.% silver nitrate, respectively), as expected, slightly differ from the energies for τ, which confirms the found proportionality between the DC conductivity and time relaxation in Equation (8), as well as the idea of thermally activated DC conductivity proposed in the Dyre model.

The resulting frequency relations of AC conductivity count in favor of the hopping mechanism of charge transfer. However, the average distance between the particles estimated from the results of high-resolution transmission electron microscopy (HRTEM) is too wide (~14–21 nm) for the process of electron tunneling to be between the inclusions of metal over a barrier whose height is of the order of 3.6 eV. Thus, the hopping mechanism of charge transfer cannot be explained only by the presence of the nanoparticles observed by HRTEM. Nevertheless, at low frequencies, the conductivity of Ag/PAN nanocomposites is several orders of magnitude lower than that of pure polymer (Figure 6a). At all temperatures of measurement, this may be seen in the frequency dependences. As a result, we may conclude that the conductivity of nanocomposites at low frequencies is still mostly attributable to electron transmission through silver nanoparticles. The cause for this is thought to be indirect electron tunneling between “big” particles via intermediate localized states, which could be related to the presence of finely scattered and atomic metallic phases in the composite [50], which cannot be observed in the HRTEM experiments. It was determined [51,52] that in the attendance of a glut of Ag^+^ ions, clusters emerge with a positive charge, which comprise silver atoms and ions. Under the dissociation of AgNO_3_ in polyacrylonitrile, we also have an excess of ions. Silver clusters in aqueous solutions were obtained and studied by an optical method [53]. In the same work, the mechanism of formation of the so-called “magic” clusters (Ag_4_^2+^, Ag_8_^2+^, and, possibly, Ag_14_^2+^) was examined along with the reasons for their stability. Ag_4_^2+^ clusters steadied by tryptophan were also investigated [54]. Accordingly, we can assume that similar clusters of silver are formed in our samples during the polymerization of acrylonitrile.

According to the theory presented in [53], Ag_14_^2+^ clusters should exhibit broad optical absorption without pronounced maxima in the region of 380–450 nm. In our samples, near this region, only an absorption band is observed, which is associated with surface plasmon resonance from silver nanoparticles (Figure 9). Therefore, it can be assumed that either there are no Ag_14_^2+^ clusters in the polymer, or their concentration is low. Ag42+ clusters give absorption peaks at 265 nm, while Ag_8_^2+^ clusters give peaks at 290 and 325 nm [52]. In the silver/PAN nanocomposite spectrum, weak minima were found at 280 and 325 nm, apparently associated with the presence of Ag_8_^2+^ clusters (Figure 9). On the contrary, these minima were not found in pure polyacrylonitrile. The absence of a minimum from Ag_4_^2+^ can be explained by significant absorption of light by the polymer matrix itself in this wavelength region. Thus, one can expect the existence of Ag_4_^2+^ and Ag_8_^2+^ clusters in our nanocomposites. Assuming that after the dissociation of AgNO_3_ in the monomer, all Ag^+^ cations during UV polymerization participate in the formation of either a metallic phase or such clusters and do not combine back with NO^3–^ anions, and the weight of the resulting nanocomposite is equal to the weight of the starting reaction mixture; we can estimate the volume fraction of silver (*η*) in PAN as follows:*η* = *V*_Ag_/*V* = *m*_Ag_ *ρ*/*m ρ*_Ag_ = *μ ρ*/*ρ*_Ag_,(9)
where *V*_Ag_ and *m*_Ag_ are the volume and weight of all Ag inclusions; *V*, *m*, and *ρ* are the volume, weight, and density of the nanocomposite film; *ρ*_Ag_ is the bulk Ag density; and *μ* is the weight fraction of Ag found from the concentration (weight fraction) of AgNO_3_ in the reaction solution. On the one hand, for the film obtained at 10 wt.% of AgNO_3_ and 15 wt.% of photoinitiator, the volume fraction according to Formula 9 is approximately 0.58%. On the other hand, *η* for the same sample, determined from the TEM and HRTEM results, was 0.16% and 0.05%, additionally. This means that the full fraction of detected nanoparticles was approximately 0.21%. Comparing the experimental and estimated values of *η*, it can be concluded that the PAN matrix may contain even smaller silver inclusions with a volume fraction of 0.37%. If the calculated 0.37% volume fraction is attributed to the Ag_4_^2+^ and Ag_8_^2+^ clusters, the distance between them may be determined. Taking into account that according to the data reported in [53] the Ag_8_^2+^ clusters form a simple cubic lattice of silver atoms, devoid of two electrons, and knowing the values of the radius of the Ag atom and the parameter of the lattice of bulk silver, the size of such clusters was estimated to be approximately equal to 1 nm (for Ag_8_^2+^). From the size and volume fraction of such silver inclusions, their density in polyacrylonitrile may be calculated according to the following equation:*δ* = 6*η*/π*d*^3^,(10)

By analogy with the determination of the distance between nanoparticles [55], the arrangement of silver clusters of size *d* in the polymer matrix may be presented in the form of a simple cubic lattice with the constant *s* + *d*. Then the average distance between such clusters (*s*) is defined as:*s* = 1/*δ*^1/3^ − *d*.(11)

For the Ag/PAN film, prepared by UV-polymerization of solution of 10 wt.% of AgNO_3_ and 15 wt.% of PI in acrylonitrile, the density of silver clusters is approximately equal to 7.0 × 10^18^ cm^−3^, and the distance between them, according to Equation (11), is about 5 nm. This is sufficient for electron tunneling.

It should be noted that when the content of AgNO_3_ in the starting mixture grew, the conductivity and dielectric constant of the obtained Ag/PAN nanocomposites both grew. Based on the results of TEM, and the reasoning concerning the presence of Ag_4_^2+^ и Ag_8_^2+^ clusters in the polymer, the volume fraction of Ag magnification as the concentration of AgNO_3_ in the initial mixture rose. The observed correlation between the metal’s volume fraction with conductivity and dielectric constant is rather consistent with various theories of effective media [12]. In this case, by growing the concentration of AgNO_3_ in the reaction mixture, we, apparently, increased not only the volume fraction of the metal but also the density of Ag_4_^2+^ and Ag_8_^2+^ clusters in the final Ag/PAN films, which provided intermediate localized states for electrons. In this case, this assumption explains the decrease in the activation energy value with an increase in the content of the silver precursor in the initial mixture [56,57,58,59,60].

## 4. Conclusions

The analysis of the frequency relations of the electrical modulus reveals the relaxation nature of Ag/PAN nanocomposites, which relies on the Maxwell–Wagner effect (interfacial polarization). The dependence of the imaginary part of the electrical modulus clearly shows relaxation maxima associated with the presence of silver nanoparticles in the polymer.

The resulting frequency dependences of AC conductivity suggest the hopping mechanism of charge transport in Ag/PAN nanocomposites. According to the transmission spectra of nanocomposites obtained at a low concentration of the photoinitiator in the polymer matrix, in addition to silver nanoparticles detected by TEM, the presence of silver clusters Ag_4_^2+^ and Ag_8_^2+^ were to be expected. It is assumed that charge transfer occurred by means of non-direct tunneling of electrons between the “large” particles due to intermediate localized states that may be associated with the attendance of Ag_4_^2+^ и Ag_8_^2+^ clusters in the polymer. Both the Dyre model and the Cole–Davidson model correspond well with the experimental results. An increase in the volume fraction of silver nanoparticles led to an increase in the conductivity of nanocomposite films, as well as a decrease in the hopping activation energy. The polymer matrix determined the electrical characteristics of Ag/PAN nanocomposites in the high-frequency band.

## Figures and Tables

**Figure 1 polymers-13-03251-f001:**
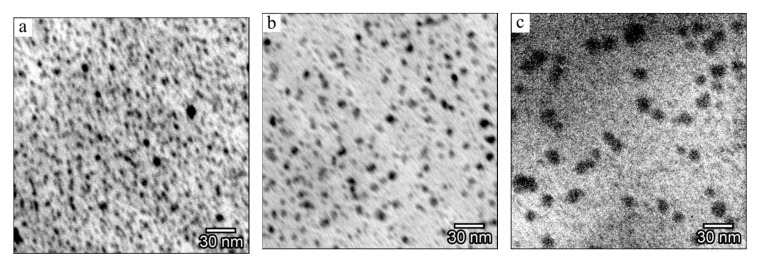
TEM images (**a**–**c**) and particle distribution in size (**d**–**f**) of Ag/PAN nanocomposite films, obtained at different concentrations of AgNO_3_, wt.%: 10 (**a**,**d**), 20 (**b**,**e**), and 30 (**c**,**f**).

**Figure 2 polymers-13-03251-f002:**
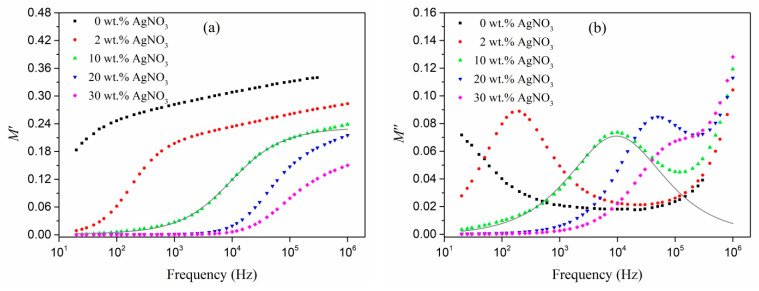
Electrical modulus vs. AC frequency at T = 60 °C: real (**a**) and imaginary (**b**) elements in Ag/PAN nanocomposites made from a mixture with various concentrations of AgNO_3_. The solid lines represent approximating curves according to the Cole–Davidson model.

**Figure 3 polymers-13-03251-f003:**
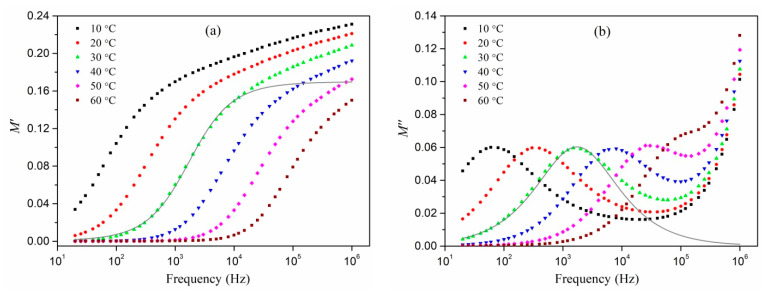
Electrical modulus vs. AC frequency: real (**a**) and imaginary (**b**) elements in Ag/PAN nanocomposite made from a mixture with 20 wt.% AgNO_3_ and 15 wt.% PI at various temperatures of measurements. The solid lines represent approximating curves according to the Cole–Davidson model.

**Figure 4 polymers-13-03251-f004:**
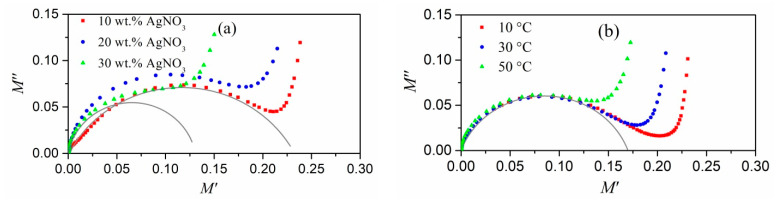
The plot of the Cole–Cole equation at *T* = 60 °C for Ag/PAN nanocomposites prepared from a mixture with different concentrations of silver nitrate (**a**) and for a Ag/PAN nanocomposite prepared from a mixture with 20 wt.% of AgNO_3_ and 15 wt.% of PI at different temperatures of measurement (**b**). The solid lines represent approximating curves according to the Cole–Davidson model.

**Figure 5 polymers-13-03251-f005:**
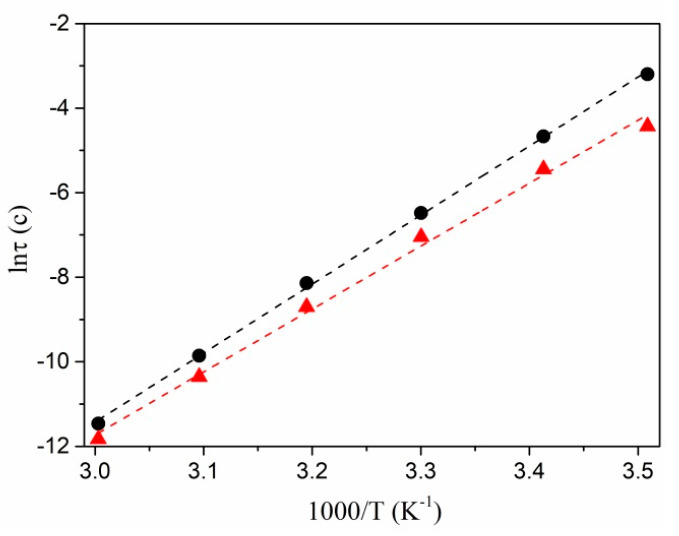
Logarithm of relaxation time vs. 1/*T* for Ag/PAN nanocomposites made at 20 (●) and 30 (▲) wt.% of AgNO_3_.

**Figure 6 polymers-13-03251-f006:**
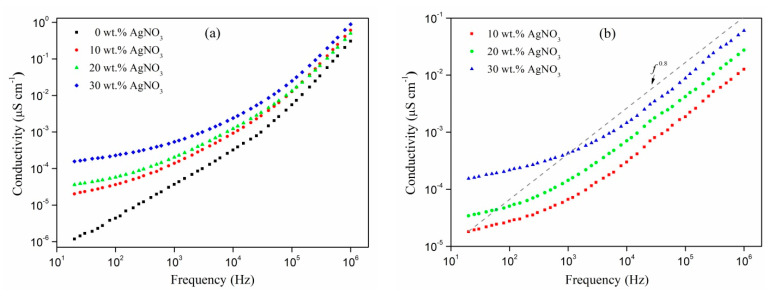
Conductivity vs. AC frequency plot for Ag/PAN nanocomposites prepared at various concentrations of silver nitrate in a starting mixture (*T* = 10 °C, PI concentration is 15 wt.%) with the account of the polymer conductivity (**a**) and without the account of the polymer conductivity (**b**). The dotted line shows the dependence of *f*^0.8^ for an arbitrary value of the coefficient *A*.

**Figure 7 polymers-13-03251-f007:**
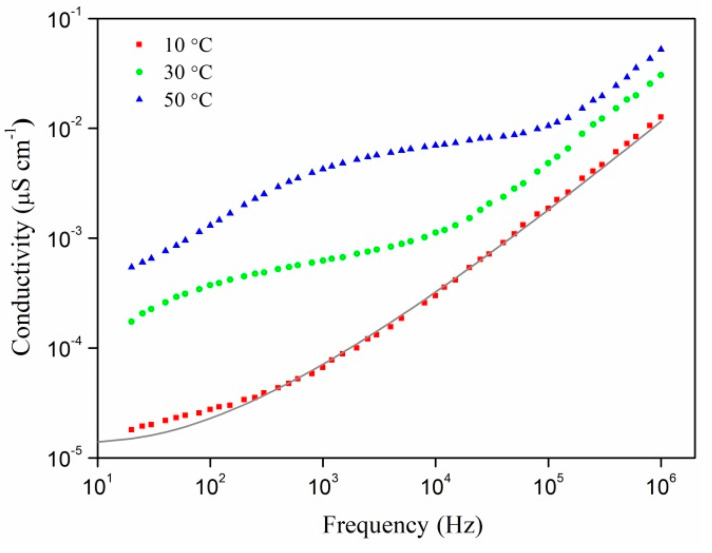
Conductivity vs. AC frequency plot for Ag/PAN nanocomposites prepared at 10 wt.% of AgNO_3_ and 15 wt.% of PI without an account of the polymer conductivity at different temperatures of measurements. The solid line shows the theoretical curve according to the Dyre model.

**Figure 8 polymers-13-03251-f008:**
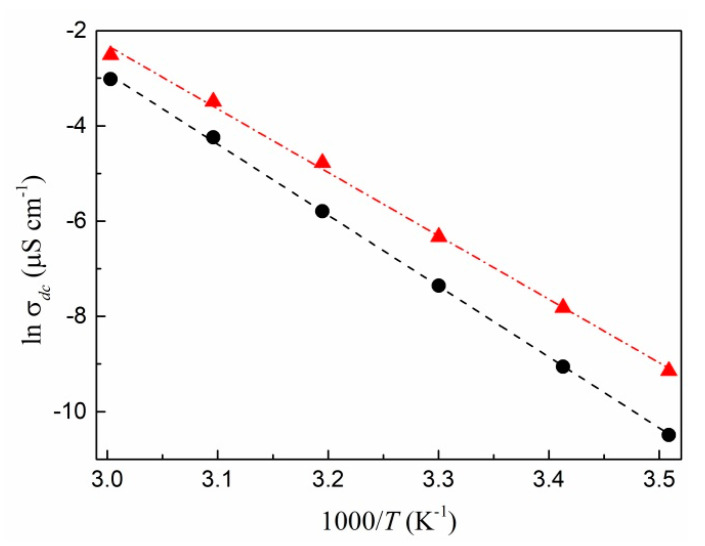
DC conductivity logarithm versus 1/*T* for Ag/PAN nanocomposite films, prepared at 20 (●) and 30 (▲) wt.% AgNO_3_.

**Figure 9 polymers-13-03251-f009:**
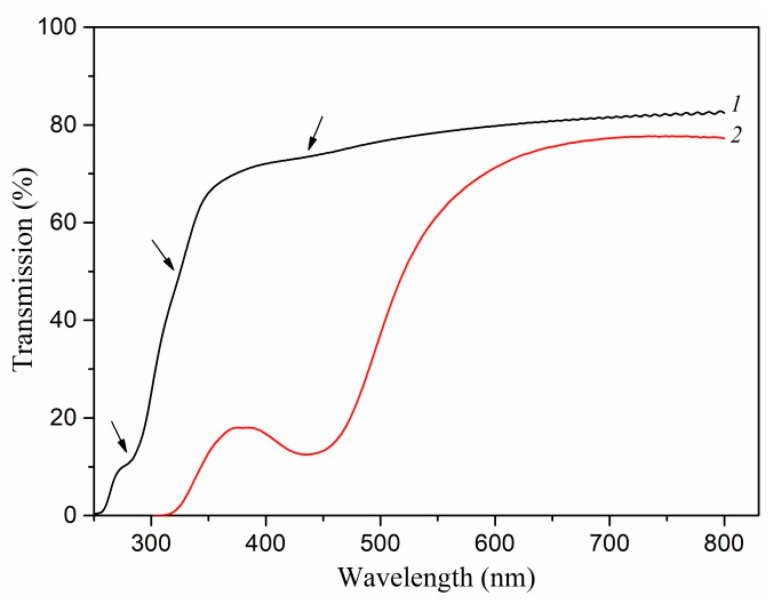
Transmission spectra of Ag/PAN nanocomposites, obtained at: 1–40 wt.% AgNO_3_ and 0 wt.% PI, 2–20 wt.% AgNO_3_ and 15 wt.% PI.

## Data Availability

The datasets generated during and/or analysed during the current study are available from the corresponding author on reasonable request.

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
