# Peer review of "Hopping Conductivity and Dielectric Relaxations in Ag/PAN Nanocomposites"

_polymers, 2021, doi:10.3390/polym13193251_

Round 1

Reviewer 1 Report

Authors presents analysis of dependence of the conductivity and electric modulus of silver nitrate/polyacrylonitrile nanocomposites on the frequency at different temperatures and proposed mechanism explaining their conductivity. Experimental part is well written and results are well discussed. But form the practically point of view, this kind of composites with such high nanoparticles loading (up to 30 wt .%) are unlikely to find practical application due to the poor possible mechanical parameters of the material..

1) In introduction Authors should explain in more detail why they chose this conductive nanofiller. Do exist other indications for their choosing as nanofiller in PAN ? For example their antibacterial efficacy due to presence sliver ions, or other reasons which could explain originality of the presented work.

I recommend accept presented work after minor revision.

Author Response

Reviewer 1:

Comments and Suggestions for Authors

Comment #1

Authors presents analysis of dependence of the conductivity and electric modulus of silver nitrate/polyacrylonitrile nanocomposites on the frequency at different temperatures and proposed mechanism explaining their conductivity. Experimental part is well written and results are well discussed. But form the practically point of view, this kind of composites with such high nanoparticles loading (up to 30 wt .%) are unlikely to find practical application due to the poor possible mechanical parameters of the material.

Answer #1

Thanks for the helpful comment. In this work, 30 wt% refers to the concentration of silver nitrate in the initial mixture, and not to the volume fraction of metal nanoparticles. However, an increase in the AgNO3 content in the mixture will undoubtedly lead to an increase in impurities in the polymer, which will affect the mechanical properties of polyacrylonitrile. Unfortunately, we have not carried out experiments to study the mechanical properties of nanocomposites. However, it has been qualitatively found that the resulting polymer films become more flexible. This behavior may be associated with an increase in the amount of residual acrylonitrile that did not undergo the polymerization reaction. We discussed this in our early work. [https://doi.org/10.1134/S1063784216110128].

Comment #2

1) In the introduction Authors should explain in more detail why they chose this conductive nanofiller. Do exist other indications for their choosing as nanofiller in PAN? For example their antibacterial efficacy is due to the presence of sliver ions, or other reasons which could explain the originality of the presented work.

Answer #2

Thanks for this important question. The relevant criteria for the selection of the filler, as well as the polymer matrix itself, were added to the introduction.

Reviewer 2 Report

The results presented in the work «Hopping conductivity and dielectric relaxations in Ag/PAN nanocomposites» were previously published in articles by the same authors, namely:

1. Kudryashov, M.A., Mashin, A.I., Logunov, A.A. et al. Dielectric properties of Ag/PAN nanocomposites. Tech. Phys. 59, 1012–1016 (2014). https://doi.org/10.1134/S1063784214070147

https://link.springer.com/article/10.1134/S1063784214070147

2. Kudryashov, M.A., Mashin, A.I., Logunov, A.A. et al. Frequency dependence of the electrical conductivity in Ag/PAN nanocomposites. Tech. Phys. 57, 965–970 (2012). https://doi.org/10.1134/S1063784212070134

https://link.springer.com/article/10.1134/S1063784212070134

3. Kudryashov, M.A., Mashin, A.I., Tyurin, A.S. et al. Morphology of a silver/polyacrylonitrile nanocomposite. Tech. Phys. 56, 92–96 (2011). https://doi.org/10.1134/S1063784211010154

https://link.springer.com/article/10.1134/S1063784211010154

4. Kudryashov, M.A., Mashin, A.I., Tyurin, A.S. et al. Metal-polymer composite films based on polyacrylonitrile and silver nanoparticles. Preparation and properties. J. Synch. Investig. 4, 437–441 (2010). https://doi.org/10.1134/S1027451010030134

https://link.springer.com/article/10.1134/S1027451010030134

5. M A Kudryashov et al 2021 J. Phys.: Conf. Ser. 1967 012046

https://iopscience.iop.org/article/10.1088/1742-6596/1967/1/012046/meta

At the same time, neither the generalization of the material, nor the improvement of the values of the studied parameters was carried out. This manuscript should be rejected as it stands. I recommend changing the status of the article to a review one and significantly changing its content.

Author Response

Reviewer 2:

Comments and Suggestions for Authors

The results presented in the work «Hopping conductivity and dielectric relaxations in Ag/PAN nanocomposites» were previously published in articles by the same authors, namely:

  1. Kudryashov, M.A., Mashin, A.I., Logunov, A.A. et al. Dielectric properties of Ag/PAN nanocomposites. Tech. Phys. 59, 1012–1016 (2014). https://doi.org/10.1134/S1063784214070147
  2. Kudryashov, M.A., Mashin, A.I., Logunov, A.A. et al. Frequency dependence of the electrical conductivity in Ag/PAN nanocomposites. Tech. Phys. 57, 965–970 (2012). https://doi.org/10.1134/S1063784212070134
  3. Kudryashov, M.A., Mashin, A.I., Tyurin, A.S. et al. Morphology of a silver/polyacrylonitrile nanocomposite. Tech. Phys. 56, 92–96 (2011). https://doi.org/10.1134/S1063784211010154
  4. Kudryashov, M.A., Mashin, A.I., Tyurin, A.S. et al. Metal-polymer composite films based on polyacrylonitrile and silver nanoparticles. Preparation and properties. J. Synch. Investig. 4, 437–441 (2010). https://doi.org/10.1134/S1027451010030134
  5. M A Kudryashov et al 2021 J. Phys.: Conf. Ser. 1967 012046

At the same time, neither the generalization of the material, nor the improvement of the values of the studied parameters was carried out. This manuscript should be rejected as it stands. I recommend changing the status of the article to a review one and significantly changing its content.

Answer

In this work, we generalized the results on the morphology of silver nanoparticles in polyacrylonitrile and electrical measurements of nanocomposite films. It is shown that the proposed Dyre (for conductivity) and Cole-Davidson (for electrical module) models are consistent with each other with respect to the obtained relaxation times and activation energies.

Reviewer 3 Report

Authors in a manuscript entitled "Hopping conductivity and dielectric relaxations in Ag/PAN nanocomposites" present interesting work about electrical conductivity and dielectric properties of Ag/PAN nanocomposites. Generally, the paper is well written and may be wanted by researchers and engineers. However, the form of presentation results should be improved. Authors should present and discuss results for all prepared samples (mentioned in manuscript) or justify why some samples are discussed or not (it appears from the manuscript that the authors prepared samples in the range of 2-40 wt.% but in different parts of the manuscript they present results for different fractions).
Therefore, I recommend to accept the manuscript after minor revision. Additionally, below are listed some issues which require consideration:
1. Sample preparation should be described in more detail, additionally, authors should give information about the range of a mass fraction of obtained nanocomposites.
2. What was the purpose of using glass plates with a conductive layer?
3. Figure 1 should contain a distribution of particles for each TEM image.
4. Why did Authors preset the fitted model for only chosen data (eg. Fig. 2-4)?
5. The Authors should also present results for the pure matrix.
6. The Authors, in their manuscript, refer to Figure 5a (line264), but there is no such figure, the same situation is for Fig. 6a (line368), and 7b (line 332). 
7. In line 407 Authors refer to formula 3.17, but there is no such equation in the manuscript.
8. The conclusion section should be extended and highlighted the findings of the work.

Author Response

Reviewer 3:

Comments and Suggestions for Authors

Comment #1

Authors in a manuscript entitled "Hopping conductivity and dielectric relaxations in Ag/PAN nanocomposites" present interesting work about electrical conductivity and dielectric properties of Ag/PAN nanocomposites. Generally, the paper is well written and maybe wanted by researchers and engineers. However, the form of presentation results should be improved. Authors should present and discuss results for all prepared samples (mentioned in manuscript) or justify why some samples are discussed or not (it appears from the manuscript that the authors prepared samples in the range of 2-40 wt.% but in different parts of the manuscript they present results for different fractions).

Answer #1

Thanks for mentioning this important point. The results were brought to a single structure. The first two TEM images have been replaced. The last TEM image was mistakenly assigned to a different sample.

Comment #2

  1. Sample preparation should be described in more detail, additionally, authors should give information about the range of a mass fraction of obtained nanocomposites.

Answer #2

The experimental part has been expanded. We added details of obtaining samples, as well as examining them on TEM.

Comment #3

  1. What was the purpose of using glass plates with a conductive layer?

Answer #3

A conductive layer on glasses is required as electrodes for electrical measurements. Thus, we have created cells like a flat capacitor. The required information has been added to the experimental section.

Comment #4

  1. Figure 1 should contain a distribution of particles for each TEM image.

Answer #4

In Figure 1, added particle distributions for all images.

Comment #5

  1. Why did Authors preset the fitted model for only chosen data (eg. Fig. 2-4)?

Answer #5

The Dyre and Cole-Davidson models were used for all electrical measurements. They were used to determine the relaxation times and activation energies, as well as the gamma parameter for the Cole-Davidson model. However, to make the figures readable, we have shown the characteristic curves of the models for only one case.

Comment #6

  1. The Authors should also present results for the pure matrix.

Answer #6

The results for the pure polymer matrix are presented in Figure 6 and have also been added to Figure 2. Unfortunately, for the pure polymer and for the nanocomposite obtained at 2 wt%, TEM images were not possible. Due to the very strong fragility of the films, it is impossible to obtain an ultrathin cut.

Comment #7

  1. The Authors, in their manuscript, refer to Figure 5a (line264), but there is no such figure, the same situation is for Fig. 6a (line368), and 7b (line 332).
  2. In line 407 Authors refer to formula 3.17, but there is no such equation in the manuscript.

Answer #7

Thanks you for pointing out the typos. Figure numbers and formulas have been corrected.

Comment #8

  1. The conclusion section should be extended and highlighted the findings of the work.

Answer #8

The conclusion section has been expanded.

Round 2

Reviewer 2 Report

Accept in present form